# Secretome of Hypoxic Endothelial Cells Stimulates Bone Marrow-Derived Mesenchymal Stem Cells to Enhance Alternative Activation of Macrophages

**DOI:** 10.3390/ijms21124409

**Published:** 2020-06-21

**Authors:** Kang-Ju Chou, Chih-Yang Hsu, Chien-Wei Huang, Hsin-Yu Chen, Shih-Hsiang Ou, Chien-Liang Chen, Po-Tsang Lee, Hua-Chang Fang

**Affiliations:** 1Division of Nephrology, Kaohsiung Veterans General Hospital, Kaohsiung 813779, Taiwan; kjchou@vghks.gov.tw (K.-J.C.); cyhsu@vghks.gov.tw (C.-Y.H.); cwhuang0824@vghks.gov.tw (C.-W.H.); chuy2@vghks.gov.tw (H.-Y.C.); shou@vghks.gov.tw (S.-H.O.); cclchen1@vghks.gov.tw (C.-L.C.); ptlee@vghks.gov.tw (P.-T.L.); 2Department of Medicine, School of Medicine, National Yang-Ming University, Taipei 112304, Taiwan

**Keywords:** mesenchymal stem cells, hematopoietic cell E- and L-selectin ligand, hypoxic endothelial cells, alternative macrophage polarization

## Abstract

We intended to explore the cellular interaction between mesenchymal stem cells (MSCs) and injured endothelial cells leading to macrophage alternative polarization in healing kidney ischemic reperfusion injury. In vivo, the amounts of recruited macrophages were significantly mitigated by MSCs in the injured tissues, especially in the group using hematopoietic cell E- and L-selectin ligand (HCELL)-positive MSCs. Compared to controls, MSCs also enhanced expression of CD206 and CD163, which was further enhanced by HCELL expression. In vitro, analysis of cytokines involving macrophage polarization showed IL-13 rather than IL-4 from MSCs agreed with expression of macrophage CD206 in the presence of hypoxic endothelial cells. Among them, HCELL-positive MSCs in contact with hypoxic endothelial cells produced the greatest response, which were reduced without HCELL or using a transwell to prevent cell contact. With blockade of the respective cytokine, downregulated MSCs secretion of IL-13 and CD206 expression were observed using inhibitors of IFN-γ and TNF-α, but not using those of TGF-β and NO. With IFN-γ and TNF-α, MSCs IL-13 secretion and CD206 expression were upregulated. In conclusion, hypoxia induces endothelial cells producing multiple cytokines. Among them, IFN-γ and TNF-α that stimulate MSCs to secrete IL-13 but not IL-4, leading to alternative polarization.

## 1. Introduction

Human mesenchymal stem cells (MSCs) first isolated from bone marrow are able to differentiate into tissues of mesodermal origin and have been experimented to replace damaged or degenerative tissues in terms of its multi-lineage differentiation potential [1]. Many previous studies reported MSCs did help the repair of tissue injury, including kidney ischemic reperfusion injury; however, the amounts of MSC engraftment could not explain the degree of tissue restoration. This suggests that tissue restoration is not the result of direct cell replacement by implanted MSCs [2,3,4,5,6].

In addition to multi-lineage differentiation potential, MSCs possess immune modulation effects that affect phenotypic expression of macrophages, which play an important role in innate immunity, controlling and regulating inflammation in order to eliminate invading pathogens and restore tissue damage [7]. By responding to exogenous pathogen-specific factors and endogenous tissue factors, macrophages can be recruited to the inflammatory site and transform into a proinflammatory M1 phenotype or an anti-inflammatory M2 phenotype depending on microenvironmental cues [8]. The classically activated macrophage M1 is induced by exposure to IFN-γ, while the alternatively activated macrophage M2 is induced by exposure to IL-4 or IL-13 [9]. Although MSCs might adopt pro- or anti-inflammatory properties, most experiments focused on the anti-inflammatory properties that endow MSCs with the ability to polarize macrophages into M2 phenotype, which could kick off a repair process and restore tissue damage [6,10,11,12,13,14]. These studies concluded the immune modulation effects play a major role in the healing potential of MSCs.

The exact mechanism of the polarization effect of MSCs remains uncertain, despite plenty of studies reporting this phenomenon. Therefore, it is interesting to explore how MSCs interact with the injured tissues and inflammatory cells. In our previous study of using MSCs in healing mouse ischemic reperfusion injury, we found MSCs were in close contact with inflammatory cells when homing to the injured tissues [6]. In this experiment, to enhance MSCs homing to the injured tissues, we transformed native CD44 on MSCs into a hematopoietic cell E-/L-selectin ligand (HCELL) that possesses potent E-selectin affinity [15]. Through fucosyltransferase VI transfection, MSCs were fucosylated on N-glycans of CD44 to become HCELL positive, thus interacting with E-selectin on injured endothelial cells. HCELL expression facilitated MSCs homing and their healing ability [6]. Moreover, the engagement of MSCs HCELL and injured endothelial cells E-selectin could result in firm adhesion between these cells, which not only optimizes paracrine effect but also might allow direct cell-to-cell interaction. Therefore, MSCs might interact with recruited inflammatory cells and the injured tissues through a paracrine effect and/or by direct cell-to-cell contact. Our previous study also found, in vitro, hypoxic endothelial cells might secrete IFN-γ to stimulate MCS IL-13 secretion that subsequently contributes to alternative macrophage polarization [6]. Based on these preliminary findings, we intended to answer the following questions: (1) Are cytokines other than IL-13 and IFN-γ involved in the polarization effect? (2) How does cell–cell contact impact the interaction? (3) What intracellular pathway of MSCs is responsible for the effect?

## 2. Results

### 2.1. In Vivo Alternative Macrophage Polarizing Effect of MSC in Ischemic Kidney Injury

The effect of MSCs on macrophage polarization in kidneys four days after ischemic reperfusion injury was analyzed using flow cytometry and immunostaining. Kidney tissues were obtained from mice subjected to a 25-min left renal artery occlusion. F4/80 expression was used for determining the total macrophages, whereas CD206 or CD163 expression was used to detect transformation of macrophages into an anti-inflammatory phenotype. Cell suspensions prepared from kidney cortex were analyzed using flow cytometry (Figure 1a). The percentages of F4/80-positive cells out of total DAPI-positive cells as well as CD206- and CD163-positive cells out of F4/80-positive cells were calculated. The percentages of macrophage in total kidney cells were significantly reduced by the treatment of MSCs compared to the ischemic control group, especially in the group treated with HCELL-positive MSCs. In our previous study, we demonstrated HCELL-positive cells possess a better homing and healing ability to injured tissues than HCELL-negative cells [6]. In kidney ischemic reperfusion injury treated with vehicle, macrophages in the injured kidney rarely expressed CD206 or CD163, while macrophages in the MSC-treated kidney expressed more CD206 and CD163, which was further enhanced by the presence of HCELL.

Immunostaining of kidney cortex sections with ischemic injury were performed using Alexa 594-labeled secondary antibodies (red) against primary anti-F4/80 antibodies and Alexa 488-labeled secondary antibodies (green) against primary anti-CD206 antibodies. Representative photomicrographs show CD206-positive cells appeared mainly in the MSC-treated mice, especially in the HCELL-positive group (Figure 1b). The results agree with those of flow cytometry.

### 2.2. In Vitro Interaction Between MSCs and Endothelial Cells Orchestrates Alternative Macrophage Polarization

In vitro tests were performed to confirm the findings of in vivo experiments and to examine the cytokine profile associated with alternative macrophage polarization. Immunostaining of murine macrophages cultivated in the media obtained from co-culture of endothelial cells and MSCs in different settings showed exclusively F4/80-positive cells in the controls and the settings of intact endothelial cells, whereas significant alternative polarization indicated by the presence of CD206-positive cells developed in the settings of co-culture of hypoxic endothelial cells and MSCs, which was more prominent in the presence of HCELL (Figure 2a). Flow cytometry was used to quantitatively assess alternative polarization, showing similar results (the top panel of Figure 2b), whereas the expression of CD86 that indicates M1 macrophage activation agreed with the settings with hypoxic endothelial cells (the bottom panel of Figure 2b). Nonetheless, the alternative polarization effect of MSCs was weakened in the absence of HCELL or using a transwell to prevent cell contact. Cytokine profiles examined using ELISA are shown in Figure 2c, indicating that hypoxia could trigger secretion of IFN-γ, TNF-α, TGF-β1, and NO from endothelial cells (the lower four panels of Figure 2c), agreeing with macrophage CD86 expression (the bottom panel of Figure 2b). Furthermore, the secretion of these cytokines was mitigated by MSCs, which was more remarkable in the setting of HCELL-positive MSCs with no transwell to prevent cell contact. However, these cytokines did not contribute to alternative macrophage polarization in the absence of MSCs. In the experimental settings with hypoxic endothelial cells, MSCs produced IL-13 but not IL-4 (the upper two panels of Figure 2c). This effect was enhanced in the setting of HCELL-positive MSCs with no transwell that allowed a direct contact of these two cells. In this experiment, an interesting finding worth emphasizing is that the interaction between hypoxic endothelial cells and HCELL-positive MSCs reduced the secretion of cytokines from hypoxic endothelial cells as compared to their non-transwell and HCELL-negative counterparts, even if it induced the greatest IL-13 secretion and alternative macrophage polarization. A possible explanation is that there might be a reciprocal proinflammatory versus anti-inflammatory cell-to-cell interaction. The greatest anti-inflammatory effect induced by these two cells might give a negative feedback to influence inflammatory cytokines secretion of hypoxic endothelial cells. Taken together, hypoxia could trigger the interaction between endothelial cells and MSCs, orchestrating a cytokine profile that favors alternative macrophage polarization. Moreover, HCELL might provide these cells with a firm contact that promotes an optimal paracrine interaction.

### 2.3. Effect of Hypoxic Endothelial Secretome and their Blockade on MSC-Induced Alternative Macrophage Polarization

IFN-γ, TNF-α, TGF-β1, and NO secreted by hypoxic endothelial cells were blocked using the respective inhibitors to examine how these cytokines contribute to activating MSCs that subsequently promoted macrophage alternative polarization. The expression of CD206 on mouse macrophages was assessed using flow cytometry in the presence of inhibitory antibodies against IFN-γ receptor, R-7050 (inhibitor of TNF-α receptor), LY364947 (ATP-competitive TGF-β1 receptor kinase inhibitor), or DL-a-lipoic acid (non-specific free radical scavenger) in a dose-dependent manner. Figure 3a shows the CD206 expression of murine macrophages as well as the IL-13 secreted from activated MSCs was mitigated in the presence of the inhibitors of IFN-γ and TNF-α but not TGF-β1 and NO. It suggests the former two cytokines participate in stimulating MSCs in favor of alternative macrophages polarization.

To assess the effect of hypoxic endothelial secretome on MSC-induced alternative macrophage polarization, flow cytometry to measure the percentage of CD206 expressing macrophages as well as IL-13 secretion again indicated that IFN-γ and TNF-α rather than TNF-β and NO produced by hypoxic endothelial cells could enhance alternative macrophage polarization, causing a dose-dependent increase of CD206-positive macrophages and IL-13 secretion (Figure 3b). Under the stimulation of IFN-γ and TNF-α, the IL-13 secreted from HCELL-positive MSCs was able to alternatively polarize macrophages. To examine the stimulating effect of NO on MSCs, L-arginine was used as substrate of endothelial NO synthase. Using 10 and 100 μM L-arginine, the concentration of NO in the presence of intact endothelial cells increased from 0.4 ± 0.1 to 6.6 ± 0.4 and 8.3 ± 0.9 μM, respectively, similar to those with hypoxic endothelial cells and selected for experiments. The results indicate NO could not stimulate MSCs to secrete IL-13 and help alternatively polarize macrophages.

In this experiment, we did not remove blockers in conditioned media when culturing mouse macrophages. Therefore, these blockers might have some direct impact on macrophage polarization. However, in the stimulation tests, the results still agree with those of blockade tests under these limitations.

### 2.4. Impact of Firm Adhesion Blockade on MSC-Induced Alternative Macrophage Polarization

Impact of firm adhesion blockade on MSC-induced alternative macrophage polarization was examined. Endothelial cells subjected to hypoxia for 4 h were co-cultured with HCELL-positive MSCs in the presence of antibodies against E-selectin, VLA-4, or VCAM-1, which are responsible for firm adhesion between these two cells. The expression of CD206 on mouse macrophages that are constitutively F4/80 positive was assessed using flow cytometry in the presence of the respective antibody in a dose-dependent manner. Figure 4 shows the expression of CD206 of macrophages as well as the IL-13 secreted from activated MSCs. All of the parameters were mitigated to a varied degree in the presence of inhibitory antibodies. The findings suggest hypoxic endothelial cells could prime MSCs into anti-inflammatory phenotype via direct cell-to-cell interaction through firm adhesion as well as paracrine effect.

### 2.5. Hypoxic Endothelial Cells Activate the MSCs Intracellular Signaling Pathways of IFN-γ and TNF-α

The MSCs intracellular signaling pathway activity of IFN-γ (Figure 5a) and TNF-α (Figure 5b) was examined using Western blotting of p-STAT1 and NF-κB respectively. With conditioned culture media from hypoxic endothelial cells, the expression of p-STAT1 was increased in HCELL-positive MSCs compared to that grown in conditioned culture media without hypoxia. This upregulation of p-STAT1 was significantly mitigated by the pathway inhibitors, either IFN-γR Ab or CYT387, in a dose-dependent manner. Furthermore, the p-STAT1 expression could be stimulated using recombinant IFN-γ, although not as prominent as using hypoxic conditioned culture media. The MSC IL-13 secretion well correlated to the changes of p-STAT1 expression. Instead of p-STAT1, the expression of NF-κB was examined to evaluate the MSCs intracellular signaling pathway activity of TNF-α, showing that the results are similar to those of IFN-γ. Taken together, the above results support our previous findings that IFN-γ and TNF-α secretion from hypoxic endothelial cells might activate MSCs secretion of IL-13, leading to alternative macrophage polarization.

## 3. Discussion

According to Mantovani et al., alternatively activated macrophages can be further differentiated into M2a, M2b, M2c, and M2d based on their activators, markers expression, and cytokines secretion [16]. In our study, we found both CD206 and CD163 expressions were significantly activated in the ischemic kidney section treated with MSCs, and IL-13, the activator of M2a subset, was secreted from MSCs in line with the CD206 expression of macrophages treated with media from coculture of hypoxic endothelial cells and MSCs [17,18,19]. In addition, the rescuing effect of MSCs on kidney function impairment agreed with the CD206 expression on ischemic reperfusion kidney injury in our previous study [6]. These indicate M2a subset might be one of the major components for tissue repair in the murine model of kidney ischemic reperfusion injury. However, we did not examine markers of other M2 subtypes, so other subsets might also be involved [19].

Understanding the mechanism and factors that influence immunosuppressive potentials of MSCs is important for their clinical application. Proinflammatory cytokines abundant in the environment of inflammation are reasonable candidates to have a pivotal role in MSC immunosuppressive function. Among them, IFN-γ secreted from T lymphocytes was demonstrated to activate immunosuppressive function of MSCs by Krampera et al. [20]. Based on this finding, several investigators assumed the strategy of IFN-γ boosting MCS immunosuppression, showing IFN-γ primed MSCs were more effective than their non-IFN-γ primed counterparts [21,22]. Ren et al. reported immunosuppressive function of MSCs could be activated in the presence of IFN-γ in combination with any of the following proinflammatory cytokines, TNF-α, IL-1α, or IL-1β [23]. However, these cytokines could not work alone without IFN-γ. IFN-γ and TNF-α were among the cytokines secreted from hypoxic endothelial cells in our study, which can account for macrophage alternatively activated property of MSCs according to Ren et al. Our results agree with those of the aforementioned studies with the exception that IFN-γ or TNF-α alone could activate MSC immunosuppression (Figure 3b).

Using inhibitors of the TGF-β1 and NO, the MSCs secretion of IL-13 and macrophage CD206 expression could not be downregulated, and the stimulation test using TGF-β1 and L-arginine, a NO substrate, did not increase IL-13 secretion and CD206 expression. Therefore, TGF-β1 and NO were also found to increase secretion from hypoxic endothelial cells, which however was not considered contributing to alternatively activating MSCs into anti-inflammatory phenotype in our study.

In our previous study, HCELL expression could enhance MSC homing and its repair ability [6]. In this study, we further demonstrated HCELL can booster polarizing effect of MSCs on macrophages. Despite lacking chemokine signaling molecules, MSCs were demonstrated to develop firm adhesion to endothelial cells via activating Rac1/Rap1 GTPase signaling pathway and subsequent association of VLA-4 and VCAM-1 [24]. The engagement of E-selectin and HCELL allows MSC rolling over endothelial cell surface against shearing force of blood flow, which might also lead to activation of other signaling pathway in MSCs, resulting in secreting more cytokines. Another explanation is that the association of HCELL and E-selectin can bring both cells in close contact, which enhances paracrine cytokines such as IFN-γ to influence MSCs intensely due to proximity.

Based on the microenvironmental cues of tissue injury, MSCs might play a pivotal role in preserving tissue integrity during invasion of pathogens or tissue injury via appropriately and timely switching of inflammatory processes between pro- and anti-inflammatory modes. Factors that have been reported to influence the switch between pro- or anti-inflammatory property include duration after injury, level of inflammatory cytokines, and type of Toll-like receptors ligation [23,25,26,27]. MSCs with proinflammatory property might be beneficial for the early stage of inflammation to boost proinflammatory response and help restrain pathogen or eliminate damaged tissue. In our study, MSCs with anti-inflammatory property rather than proinflammatory property were observed either in kidney tissue four days after ischemic injury or in cell culture subjected to 4-h hypoxic insult. Accordingly, the exclusive MSC phenotype expression of anti-inflammatory property could be explained by the timing when to evaluate the property of MSCs or the mode of injury used in our experiments. Therefore, when performing cell-based therapy using MSCs, the timing of administration and disease-associated inflammation status should be taken into consideration.

Several previous studies reported MSCs function to repair kidney injury through alternatively activated macrophages, an immune modulation effect [6,11,13]. The present study revealed a possible scheme for this immune modulation effect in the murine model triggered by hypoxic endothelial cells (Figure 6). Briefly, hypoxic tissue damage might stimulate endothelial cells to secrete various cytokines, especially IFN-γ and TNF-α that activate homing MSCs through their respective receptor and downstream signaling. Activated MSCs then secrete IL-13 but not IL-4 to induce alternative activation of M2a macrophages. The canonical cytokines of type 2 immune response, IL-4 and IL-13, involving helminth infection and allergic disorders, are the most important cytokines responsible for inducing alternative activation of macrophages [17,18]. These cytokines act through receptor complexes on macrophages to stimulate downstream signaling, which subsequently increases STAT6 and its associated transcription factors [28].

In conclusion, hypoxia induces endothelial cells producing multiple paracrine stimuli, including IFN-γ, TNF-α, TGF-β1, and NO. Among them, IFN-γ in combination with TNF-α could stimulate MSCs to secrete IL-13, leading to macrophage alternative polarization.

## 4. Materials and Methods

### 4.1. Animals, Cells, and Condition Medium Preparations

C57BL/6 mice of 25–30 g were from the National Laboratory Animal Breeding and Research Center, Taipei, Taiwan. The study protocol was approved by the Institutional Animal Care and Use Committee of Kaohsiung Veterans General Hospital (2016-A078, 31/05/2015). To induce ischemic reperfusion injury, mice were anesthetized and vascular clamps were applied across the left renal pedicle for 25 min. The mice were then injected intravenously with saline or MSCs. Kidneys were harvested for flow cytometry analysis and immunostaining 4 days after injury.

To obtain MSCs, mice were killed and femurs were aseptically removed. Bone marrow was flushed from the shaft with DMEM medium (Sigma-Aldrich Corp., St. Louis, MO, USA) containing 5% FCS (Invitrogen, Waltham, MA, USA) plus penicillin/streptomycin (100 U/mL to 0.1 mg/mL; Invitrogen) and then filtered through a 100-µm sterile filter (Falcon) to produce a single-cell suspension. After adding 10% FCS, filtered bone marrow cells were allowed to adhere for 6 h. Non-adherent cells were then removed. After 2–3 weeks, adherent cells were detached by trypsin-EDTA (0.5 to 0.2 g/L; Invitrogen), washed with PBS, and used for experiments at passage 8–10.

Human fucosyl transferrase VI plasmid cDNA containing 2 copies integrated pSV2neo DNA with G418 resistance was purchased from OriGene (OriGene Technologies, Rockville, MD, USA). To endow MSCs with HCELL, MSCs were transfected using liposome method with empty vectors as controls according to the manufacturer’s instructions. After transfection, MSCs were examined to verify the presence of HCELL using flow cytometry and their mesenchymal potential to differentiate toward osteoblasts and adipocytes as previously described [6].

For in vitro studies, mouse microvascular endothelial cells (MMEC) and macrophages (RAW 264.7) from ATCC were used. For all experiments involving hypoxia, mouse endothelial cells of 2 × 10^5^ in 1 mL DMEM without glucose were subjected to a hypoxic condition with 1% O_2_, 5% CO_2_, and 94% N2 for 4 h using a hypoxic chamber (NexBiOxy Hypoxia System, NEXBIOXY Inc., Hsinchu, Taiwan). Then, MSCs of 1 × 10^5^ in 3 mL DMEM with low glucose were added and cocultured for 24 h to obtain conditioned media for cytokines measurement or mouse macrophage culture. If examining blockade of endothelial secretome, blockers were added in the coculture. To assess macrophage polarization, mouse macrophages of 1 × 10^6^ in 1 mL DMEM with high glucose were then mixed with condition media 1 mL and cultured for 24 h before assays.

### 4.2. Analysis of Macrophage Phenotype Using Flow Cytometry

Cell suspensions prepared from harvested kidneys and mouse macrophages culture using conditioned media obtained from the depicted experimental settings were examined using flow cytometry. Conditioned media were obtained using coculture of MSCs and mouse microvascular endothelial cells on different experimental conditions. For flow cytometry, approximately 2 × 10^5^ cells were incubated on ice for 30 min, and resuspended in 100 μL of PBS. Cells were first incubated with primary antibodies against F4/80 (Cell Signaling, Danvers, MA, USA), CD86 (Abcam, Cambridge, UK), CD206 (Abcam), or CD163 (Abcam) for 30 min at 4 °C in the dark, followed by appropriate secondary antibodies labeled with Alexa 594 or Alexa 488. The labeled cells were then washed and resuspended in PBS for analysis. As negative controls, isotype-matched antibodies with the corresponding fluorescent labeling were used. Cells were examined by FACS Callibur cytometry (BD Biosciences, Franklin Lakes, NJ, USA) and analyzed using WinMDI 2.8 software (The Scripps Institute, La Jolla, CA, USA).

### 4.3. Immunostaining of Macrophage Phenotype

Kidney cortex sections or macrophages grown on coverslips using conditioned media from different experimental settings were treated with 4% paraformaldehyde in PBS for 10 min and air dried. After washing with PBS, the specimens were permeabilized with 0.4% Triton X-100 and 2% BSA in PBS for 15 min. After washing, the specimens were blocked with 10% normal goat serum for 30 min and incubated with primary antibodies against F4/80 (Cell Signaling) and CD206 (Abcam) in PBS containing 0.1% Triton X-100. After washing, Alexa 594-conjugated and Alexa 488-conjugated secondary antibodies were used for fluorescence images analysis. Prior to confocal fluorescence microscopic analysis (LSM 5 Pascal, Zeiss, Jena, Germany), the sections were mounted with a fluorescence mounting solution and stained with DAPI.

### 4.4. Measurement of Cytokines

Enzyme-linked immunosorbent assay (ELISA) was used to measure the secreted cytokines, including IL-4 and IL-13, using the mouse ELISA Ready-SET-Go kits from eBioscience as well as IFN-γ, TNF-α, and TGF-β using the Mouse uncoated ELISA kits from Invitrogen. In addition, NO was measured using the Total Nitric Oxide Assay from R&D Systems (Minneapolis, MN, USA) according to the manufacturer’s instructions. Conditioned media of coculture of MSCs and MMEC in different experimental settings were obtained for analysis. The role of direct contact of both cells in macrophage polarization was assessed using a transwell system of pore size 0.4 μm (Corning Inc., Coring, NY, USA) that prevents direct contact of these cells.

### 4.5. Blockade of Endothelial Secretome

Action of IFN-γ, TNF-α, TGF-β, and NO secreted by endothelial cells under hypoxic condition were blocked using the following preparations, respectively: inhibitory antibodies against IFN-γ receptor 1 from Sigma-Aldrich; R-7050 from Sigma-Aldrich, inhibiting TNF-α-induced binding of TNF-αRI with TNFαR-associated death domain protein and receptor interacting protein 1; LY364947 from Abcam, a potent selective ATP-competitive TGF-β1 receptor kinase inhibitor; and DL-alpha-lipoic acid from Abcam, a non-specific free radical scavenger to deplete NO.

### 4.6. Blockade of HCELL-Mediated Firm Adhesion

Cytokines from MSCs and the percentage of CD206 expression of mouse macrophages were measured to assess the impact of HCELL-mediated firm adhesion between hypoxic endothelial cells and MSCs on macrophage polarization using antibodies against E-selectin from Bioss (Woburn, MA, USA), VCAM-1 of endothelial cells, and VLA-4 of MSCs from eBioscience.

### 4.7. Western Blotting

Cell lysates of MSCs subjected to the depicted treatments in different experimental settings were obtained for analysis. The immunoblotting of p-STAT1 and NF-κB were used to assess the intracellular signaling induced by IFN-γ and TNF-α, respectively. Blockade of these pathways were performed using an inhibitor of JAK1 and JAK2 (CYT387 from Santa Cruz Biotechnology, Santa Cruz, CA, USA) for IFN-γ intracellular pathway and an IKK-2 inhibitor (benzoxathiole compound ab145954 from abcam) for TNF-α intracellular pathway. Total protein extracts were quantified using Coomasie protein assay reagent (PIERCE, Thermofisher Scientific, Waltham, MA, USA). The samples were separated on 10% SDS polyacrylamide gel and transferred to a polyvinylidene difluoride membrane (NEN Life Science, Boston, MA, USA). Membranes were blocked in skim milk 1 h at room temperature and incubated with rabbit monoclonal antibodies against p-STAT1 (Abcam) and rabbit monoclonal antibodies against NF-κB (Abcam) overnight at 4 °C. The membranes were washed and incubated with horseradish peroxidase-conjugated secondary antibodies (LEADGENE, Tainan, Taiwan) for 1 h at room temperature. The blots were washed and developed using an enhanced chemiluminescence kit (VISUAL PROTEIN, Taipei, Taiwan). Quantitative analysis normalized with β-actin was performed by conducting densitometry using Image Pro Plus 7.0 software (Media Cybernetics Inc.).

### 4.8. Statistics

Data were expressed as mean ± SEM or percentage. Nonparametric one-way ANOVA was used to examine the differences among groups. Two-tailed *p* values less than 0.05 were considered statistically significant. Statistical analyses were conducted using SPSS version 12.0.1C software (Chicago, IL, USA).

## Figures and Tables

**Figure 1 ijms-21-04409-f001:**
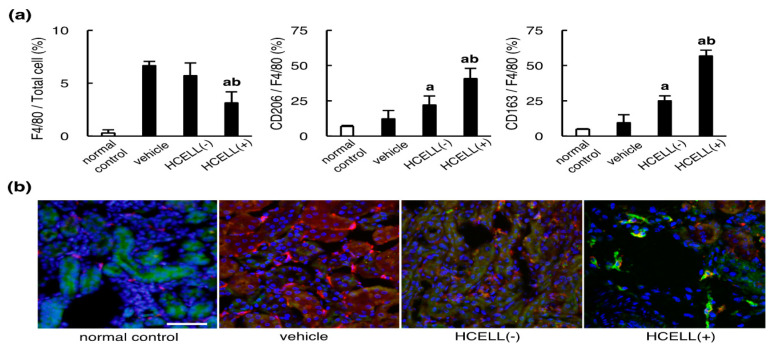
In vivo alternative macrophage polarizing effect of MSCs in ischemic kidney injury. (**a**) Cell suspensions prepared from kidney cortex with ischemic injury (black bars) and normal control (empty bar) were analyzed using flow cytometry. The percentages of F4/80-positive cells out of total cells as well as CD206- and CD163-positive cells out of F4/80-positive cells were calculated and analyzed. n = 6 in each group; a, *p* < 0.05 versus vehicle-treated group; b, *p* < 0.05 versus HCELL(-) group. Ischemic kidney injury increased macrophage numbers significantly in all analyses, black bar versus empty bar (not indicated). (**b**) Immunostaining of kidney cortex sections with ischemic injury and normal control was performed. Representative photomicrographs show CD206-positive cells (green) appeared mainly in the MSC-treated mice, especially in the HCELL(+) group, while F4/80-positive cells (red) were exclusively found in the vehicle group. Normal control shows less F4/80-positive macrophages as compared to its ischemic counterparts. Scale bar, 50 μm.

**Figure 2 ijms-21-04409-f002:**
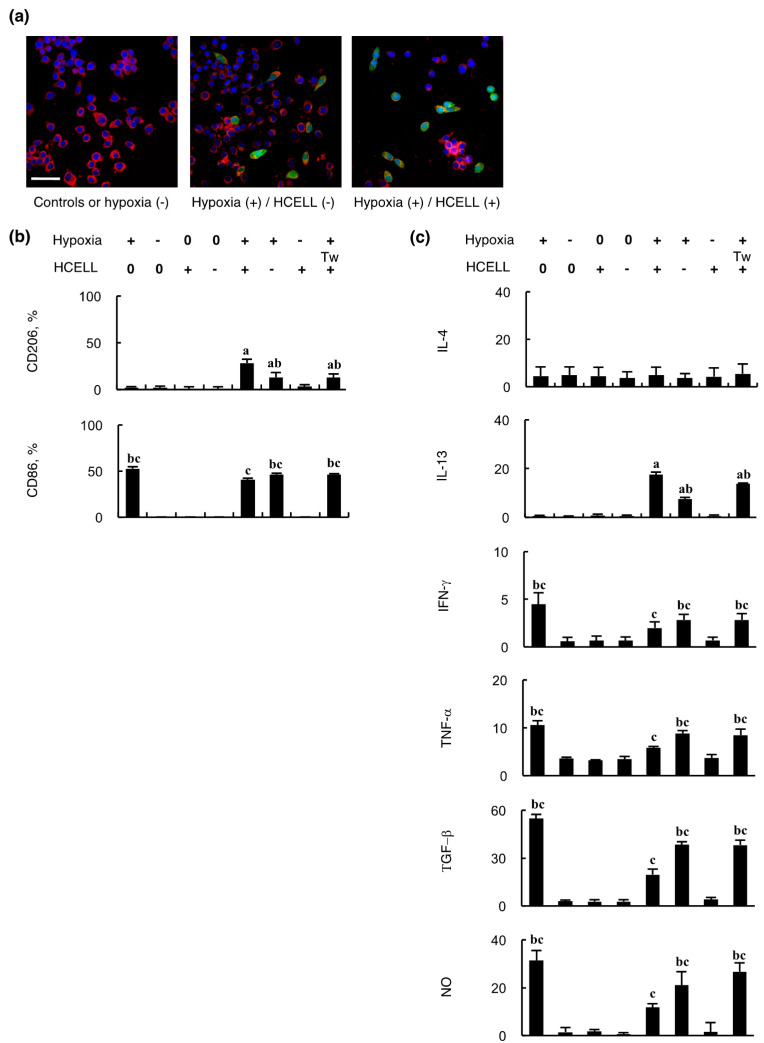
In vitro interaction between MSCs and hypoxic endothelial cells orchestrates alternative macrophage polarization. Before 24 hours’ coculture with MSCs, endothelial cells were subjected to hypoxia for 4 h. The conditioned media obtained from the depicted experimental settings were used to culture mouse macrophages for 24 h to assess alternative macrophage polarization using immunostaining (**a**) and flow cytometry (**b**). Immunostaining was performed using Alexa 594-labeled anti-F4/80 (red) and Alexa 488-labeled anti-CD206 (green). Scale bar, 50 μm. In flow cytometry, Alexa 488-labeled anti-CD86 was added to assess classical macrophage activation. (**c**) The condition media was also used to measure cytokines (pg/ 10^5^ cells/ 24 h) and NO (ng/ 10^5^ cells/ 24 h) using ELISA. (**b**,**c**) The left 4 experimental settings using single cell type were controls (0 stands for no cell). Tw stands for transwell used to prevent cell contact. Endothelial cells were either subjected to hypoxia for 4 h or not, while MSCs were endowed with HCELL as depicted. *n* = 6 in each group; a *p* < 0.05 versus controls; b *p* < 0.05 versus the setting of co-culture of HCELL-positive MSCs and hypoxic endothelial cells; c *p* < 0.05 versus settings of endothelial cells without hypoxia and controls without endothelial cells.

**Figure 3 ijms-21-04409-f003:**
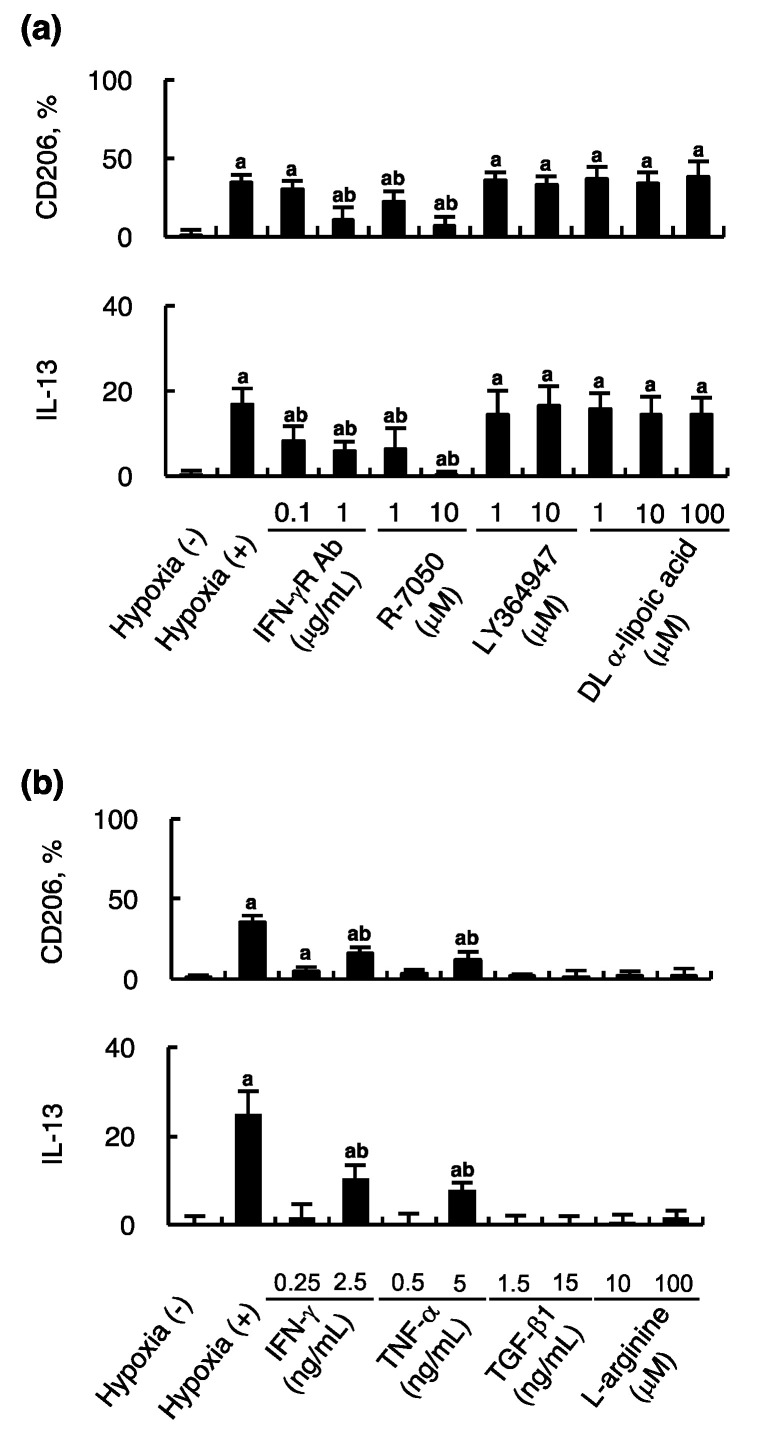
Effect of hypoxic endothelial secretome and their blockade on MSC-induced macrophage polarization. (**a**) Endothelial cells subjected to hypoxia for 4 h were co-cultured with HCELL-positive MSCs in the presence of anti-IFN-γ receptor inhibitory antibodies (IFN-γR Ab), R-7050 (inhibitor of TNF-α receptor), LY364947 (ATP-competitive TGF-β1 receptor kinase inhibitor), or DL-α-lipoic acid (non-specific free radical scavenger). Coculture of endothelial cells without hypoxia and HCELL-positive MSCs were used as controls. After 24 hours’ coculture, the conditioned culture media were then used for culturing murine macrophages for 24 h or measuring IL-13 (pg/ 10^5^ cells/ 24 h). Flow cytometry was performed to assess alternative macrophage polarization using Alexa 594-labeled anti-F4/80 and Alexa 488-labeled anti-CD206. IL-13 was measured using ELISA. n = 6 in each group; a, *p* < 0.05 versus coculture of endothelial cells without hypoxia and HCELL-positive MSCs; b, *p* < 0.05 versus coculture of endothelial cells with hypoxia and HCELL-positive MSCs. (**b**) Endothelial cells without hypoxia were co-cultured for 24 h with HCELL-positive MSCs in the presence of IFN-γ, TNF-α, TNF-β1 and L-arginine. In the absence of cytokines, intact endothelial cells were used as negative controls, whereas endothelial cells subjected to hypoxia for 4 h were used as positive controls. L-arginine was used as substrate of endothelial NO synthase and resulting NO was measured to select a proper L-arginine concentration. After 24 hours’ coculture, the conditioned culture media were then used for culturing murine macrophages for 24 h or measuring IL-13 (pg/ 10^5^ cells/ 24 h) using ELISA. Flow cytometry was performed to assess alternative macrophage polarization using Alexa 594-labeled anti-F4/80 and Alexa 488-labeled anti-CD206. *n* = 6 in each group; a, *p* < 0.05 versus negative controls without hypoxia; b, *p* < 0.05 versus positive controls with hypoxia.

**Figure 4 ijms-21-04409-f004:**
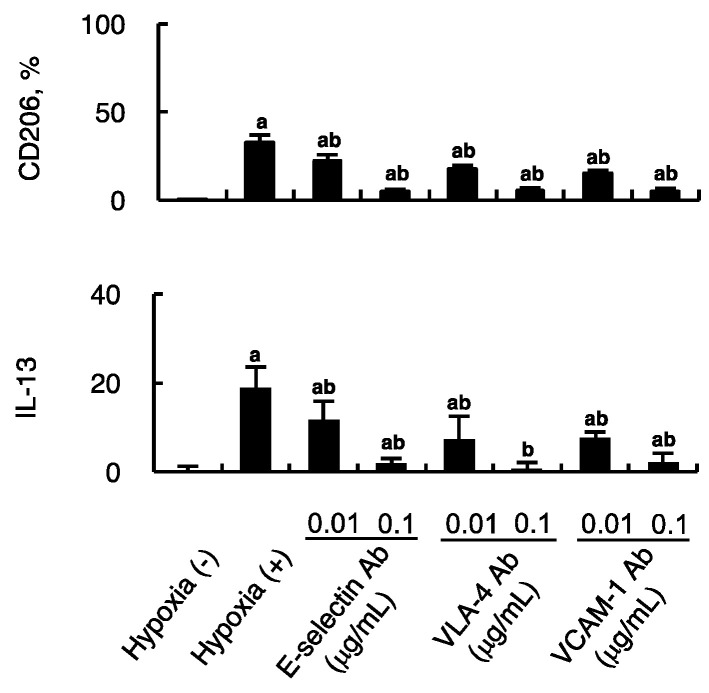
Impact of firm adhesion blockade on MSC-induced alternative macrophage polarization. Antibodies against E-selectin, VLA-4, and VCAM-1 were used to assess the effect of blockage of firm adhesion on MSC-induced macrophage polarization. Endothelial cells subjected to hypoxia for 4 h were co-cultured with HCELL-positive MSCs in the presence of antibodies against E-selectin, VLA-4, or VCAM-1. Intact endothelial cells were used as negative controls. After 24 hours’ coculture, the conditioned culture media were then used for culturing murine macrophages or measuring IL-13 (pg/ 10^5^ cells/ 24 h) using ELISA. Flow cytometry was performed to assess alternative macrophage polarization using Alexa 594-labeled anti-F4/80 and Alexa 488-labeled anti-CD206. n = 6 in each group; a, *p* < 0.05 versus negative controls; b, *p* < 0.05 versus coculture of HCELL-positive MSCs and hypoxic endothelial cells without inhibitory antibodies.

**Figure 5 ijms-21-04409-f005:**
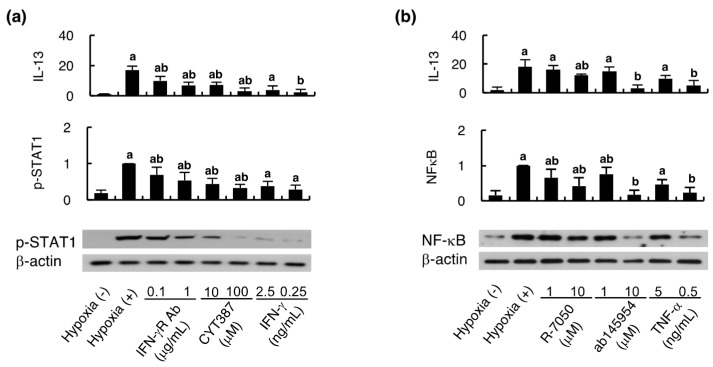
Hypoxic endothelial cells activate the MSCs intracellular signaling pathways of IFN-γ and TNF-α. The culture media obtained from endothelial cells subjected to hypoxia for 4 h were used for culturing HCELL-positive MSCs in the presence of the inhibitors of signaling pathways of IFN-γ (**a**) and TNF-α (b), whereas those from intact endothelial cells were used for assessing recombinant IFN-γ (**a**) and TNF-α (**b**) stimulation. After 24 hours’ culture in the depicted experimental settings, MSCs pellets were subjected to Western blotting to measure p-STAT1 and NF-κB for assessing IFN-γ and TNF-α pathway activation respectively, while the supernatant was used to measure IL-13 level with ELISA. The representative Western blottings are shown. The quantitative results of Western blotting were adjusted according to β-actin level. n = 8 in each group; a, *p* < 0.05 versus conditioned culture media from intact endothelial cells without stimulation; b, *p* < 0.05 versus conditioned culture media from hypoxic endothelial cells without inhibitors. (IFN-γR Ab, anti-IFN-γ receptor inhibitory antibodies; CYT387, JAK1/JAK2 inhibitor; R-7050, inhibitor of TNF-α receptor; ab145954, IKK-2 inhibitor/NF-κB activation inhibitor VI).

**Figure 6 ijms-21-04409-f006:**
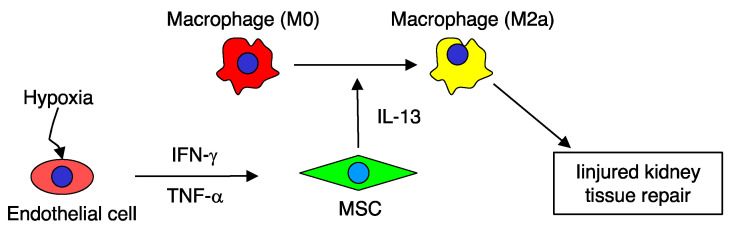
Hypoxia induces endothelial cells producing multiple paracrine stimuli, including IFN-γ and TNF-α that subsequently stimulate MSCs to secrete IL-13, leading to macrophage alternative polarization.

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
