# Peer review of "Secretome of Hypoxic Endothelial Cells Stimulates Bone Marrow-Derived Mesenchymal Stem Cells to Enhance Alternative Activation of Macrophages"

_ijms, 2020, doi:10.3390/ijms21124409_

Round 1
Reviewer 1 Report
The paper clearly improved and this reviewer is satisfied with the incorporated changes.
Reviewer 2 Report
According to me, the revision performed by the Authors is exhaustive.
Reviewer 3 Report
Authors did not provide the answers to reviewers document which complicates the evaluation of the manuscript modifications. However, they did modify the manuscript according to my comments and both results and methodology are much clearer now improving the quality of the paper.
This manuscript is a resubmission of an earlier submission. The following is a list of the peer review reports and author responses from that submission.
Round 1
Reviewer 1 Report
Chou and colleagues studied mechanisms that might underlie MSC driven polarization of macrophages. The manuscript contains several interesting elements but this reviewer has a number of concerns:
In general the article is difficult to read:
The abstract and intro would benefit from more and essential background info that is not or too late provided. For example, the renal I/R part is unclear in the abstract and is not mentioned in the intro. Providing a clinical context in the intro would be helpful.
Please be more concrete about the interaction between MSC and other cells: there is now room for misunderstanding.
The rationale to transfect MSCs is largely missing; refer to previous work is insuffcient
More info on HCELL in the intro would be very helpful.
Figures are not self explanatory and it takes a while before you understand the setup and interpretation.
With respect to the experiments, it would be important to describe the number of macrophages including the CD206 and CD163+ ones in healthy controls (figure 1).
The in vitro experiments are interesting but in general the conclusion is based on 2 or even 1 (CD206) M2 marker. The phenotype of macrophages is a continuous balance between several subtypes. Its therefore of importance to incorporate additional markers for M2 and especially provide info on M1 markers to create a better overview. In line, the statement in lines 159 and 160 should be toned down.
There are numerous studies which show that IFN-g and TNF-a are important MSC licensing molecules. This should be acknowledged and discussed.
The findings described in figure 4, on the adhesion molecules are interesting and novel but the rationale to study these 3 adhesion molecules is missing.
Minor comment:
In figures 3 till 5 please use other symbols than * to indicate significance between different conditions as in standard * stands for p<0.05, ** for p<0.01 and *** for p<0.001
Reviewer 2 Report
The paper demonstrates by in vivo and in vitro experiments that the secretome of hypoxic endothelial cells stimulates mesenchymal stem cells to secrete IL-13, leading to the activation of the anti-inflammatory M2 phenotype of macrophages, involved in the repair of tissue injury.
The paper is interesting and the experimental setting is appropriate and exhaustive, starting from in vivo evidence of the effect of stimulated-MSCs on alternative macrophage activation in ischemic kidney injury, thorough in vitro tests supporting the role of cytokines produced by hypoxic endothelial cells, in particular IFN-γ and TNF-α, to stimulate MSCs and IL-13 secretion, concluding with experiments that demonstrates the activation of IFN-γ and TNF-α signaling pathways in MSCs stimulated by hypoxic endothelial cells.
Observations:
- at line 22 of the abstract there are some typo mistakes: the Greek letters are not reported in the cytokines' names.
- Figure 2 is not immediately clear; however, it indicates that the interaction between endothelial cells and MSCs reduces the secretion of cytokines induced by hypoxia, even if it increases IL-13 secretion and alternative macrophage polarization. This concept should be better explained.
Reviewer 3 Report
The manuscript aims at analyzing the effect of the crosstalk between endothelial cells and mesenchymal stem cells on macrophages recruitment and activation. The paper includes plenty of experimental data with promising findings. However, the lack of information regarding the experimental conditions tested makes impossible to correctly evaluate the data shown. Authors keep saying on the materials and methods and the results sections, data was "obtained from the depicted experimental settings" but there is not detailed information of these conditions. There is no information at all about the conditioned media preparation. Which times points did they use to prepare this media? Did they use the same time points for all the conditions? Which cell density did they used? Did they use always the same density? How did they performed the “hypoxia” experiments? Were both cell types subjected to hypoxia or only endothelial cells (as it seems from the results section)? Did they dilute the conditioned media with fresh media or use it directly on macrophages? How did they performed the blockade of endothelial secretome? Did they add the IFN-γ, TNF-α, TGF-β, and NO “blockers” to the media during the co-culture of endothelial cells with mesenchymal stem cells? In this case, how did they remove these “blockers” before adding the media to macrophages to ensure there is no effect of the “blockers” themselves on the macrophage response?
Furthermore, authors used mesenchymal stem cells of high passages (8-10 passages). Is there any reason why they used such high passages?
The clarification of these points is crucial to further analyze the experimental data obtained.